# S-1-propenyl-L-cysteine suppresses lipopolysaccharide-induced expression of matrix metalloproteinase-1 through inhibition of tumor necrosis factor-α converting enzyme-epidermal growth factor receptor axis in human gingival fibroblasts

**Hiroshi Nango** ⬦ *, **Masahiro Ohtani**

Central Research Institute, Wakunaga Pharmaceutical Co., Ltd., Akitakata, Hiroshima, Japan

* nangou_h@wakunaga.co.jp

## Abstract

Periodontal disease is the most common dental health problem characterized by the destruction of connective tissue and the resorption of alveolar bone resulting from a chronic infection associated with pathogenic bacteria in the gingiva. Aged garlic extract has been reported to improve gingival bleeding index and probing pocket depth score in patients with mild to moderate periodontitis. Although our previous study found that aged garlic extract and its constituents suppressed the tumor necrosis factor-α-induced inflammatory responses in a human gingival epithelial cell line, the mechanism underlying the effect of aged garlic extract on the destruction of the gingiva remains unclear. The present study investigated the effect of S-1-propenyl-L-cysteine, one of the major sulfur bioactive compounds in aged garlic extract, on the lipopolysaccharide-induced expression of matrix metalloproteinases in human gingival fibroblasts HGF-1 cells. Matrix metalloproteinases are well known to be closely related to the destruction of the gingiva. We found that S-1-propenyl-L-cysteine suppressed the lipopolysaccharide-induced expression and secretion of matrix metalloproteinase-1 in HGF-1 cells. In addition, S-1-propenyl-L-cysteine inhibited the lipopolysaccharide-induced phosphorylation of epidermal growth factor receptor and expression of the active form of tumor necrosis factor-α converting enzyme. Furthermore, the inhibitors of epidermal growth factor receptor tyrosine kinase and tumor necrosis factor-α converting enzyme, AG-1478 and TAPI-1, respectively, reduced the lipopolysaccharide-induced protein level of matrix metalloproteinase-1, as did S-1-propenyl-L-cysteine. Taken together, these results suggested that S-1-propenyl-L-cysteine suppresses the lipopolysaccharide-induced expression of matrix metalloproteinase-1 through the blockade of the tumor necrosis factor-α converting enzyme-epidermal growth factor receptor axis in gingival fibroblasts.

**Data Availability Statement:** All relevant data are within the manuscript and its Supporting Information files.

**Funding:** The funder provided support in the form of salaries for authors (HN, and MO), but did not have any additional role in the study design, data collection and analysis, decision to publish, or preparation of the manuscript. The specific roles of these authors are articulated in the 'author contributions' section.

**Competing interests:** I have read the journal's policy and the authors of this manuscript have the following competing interests: Authors (H.N., and M.O.) are employed by Wakunaga Pharmaceutical Co., Ltd. and are listed in the affiliation section.

## Introduction

Gingivitis and periodontitis are the most common periodontal diseases [1]. Gingivitis is an initial inflammatory condition of the gingiva triggered by oral pathogenic bacteria in subgingival biofilms such as *Porphyromonas gingivalis*, *Tannerella forsythia*, and *Treponema denticola* [2]. When left untreated, gingivitis may advance to periodontitis characterized by the destruction of gingiva and alveolar bone, resulting in gingival recession, tooth mobility, and tooth loss [1]. Patients with periodontal diseases have a higher risk of multiple tooth loss, thereby compromising mastication and nutrient intake [3]. In addition, recent studies have focused on periodontal diseases as an important risk factor in systematic diseases including Alzheimer's disease [4], cardiovascular diseases [5], and carcinogenesis [6]. Therefore, the prevention of periodontal diseases is essential for the quality of life and overall general health.

Gingival connective tissue consists of stromal cells and extracellular matrix components that are mainly composed of fibroblasts and immune cells and collagen fiber, respectively [7, 8]. Since fibroblasts are abundant in gingival connective tissue and play an important role in maintaining the balance between the production and destruction of collagen, dysfunction of fibroblasts leads to gingival breakdown [9]. Matrix metalloproteinases (MMPs) are members of the zinc- and calcium-dependent enzyme family that degrades extracellular matrix proteins, thereby destroying the gingiva [10]. It has been reported that the levels of MMP-1, -2, -3, -8, -9, and -13 were elevated in gingival tissue specimens from patients with periodontitis [11–15]. *P. gingivalis*-derived lipopolysaccharide (LPS) also increased the mRNA levels of MMP-1, -2, and -3 in cultured human gingival fibroblasts [16, 17]. Taken together, these results suggest that MMPs secreted from inflamed gingival fibroblasts play a key role in the progression of periodontal disease through the degradation of the gingiva.

Aged garlic extract (AGE) is one of the garlic products manufactured by aging garlic (*Allium sativum* L.) in the water-ethanol mixture for more than 10 months at room temperature [18]. AGE contains several bioactive sulfur-containing amino acids, including *S*-allyl-L-cysteine, *S*-1-propenyl-L-cysteine (S1PC), and *S*-allyl-L-mercaptocysteine that possess such properties as antioxidation [19, 20], anti-inflammation [21–23], and immunomodulation [24–26]. A clinical trial on patients with mild to moderate periodontitis demonstrated that daily intake of AGE for 4 months improved gingival bleeding index [27]. In support of this result, our recent study showed that AGE and its sulfur constituents such as S1PC suppressed tumor necrosis factor-$\alpha$ (TNF-$\alpha$)-induced inflammatory responses in human gingival epithelial Ca9-22 cells [28]. In another clinical study, daily intake of AGE for 18 months reduced probing pocket depth [29], suggesting that AGE is useful for preventing the progression of periodontal disease by suppressing gingival inflammation and destruction. In the present study, we assessed the possible mechanism of AGE action on the destruction of the gingiva by examining whether S1PC, one of the major sulfur bioactive compounds in AGE, inhibits *P. gingivalis*-derived LPS-induced expression of matrix metalloproteinases in human gingival fibroblasts HGF-1 cells.

## Materials and methods

### Regents

All chemicals were purchased from FUJIFILM Wako Pure Chemical Corporation (Osaka, Japan) unless stated otherwise. AG-1478 and TAPI-1 were obtained from Cayman Chemical (Ann Arbor, MI, USA). *P. gingivalis*-derived LPS was purchased from Invitrogen (Waltham, MA, USA). The primary antibodies against EGFR (GTX121919) and pEGFR (GTX132810) were from GeneTex International Corporation (Hsinchu, Taiwan). Other primary antibodies

against MMP-1 (#54376), cleaved-ADAM17 (BS7064), and β-actin (PM053-7) were from Cell Signaling Technology (Danvers, MA, USA), Bioworld Technology (Bloomington, MN, USA) and MBL Life Science (Nagoya, Japan), respectively. The HRP-conjugated secondary antibodies against mouse (#7076S) and rabbit (#7074S) were obtained from Cell Signaling Technology.

## Preparation of S1PC

AGE was prepared as previously described [30]. S1PC was isolated and purified to >98% purity from AGE by a liquid chromatography-mass spectrometry system consisting of an Ultimate 3000 and a Q-Exactive (Thermo Scientific, Waltham, MA, USA), and by a VNMRS-500 NMR spectrometer (VARIAN Inc., Palo Alto, CA, USA) at 500 MHz and 125 MHz [31].

## Cell culture

The human gingival fibroblasts (CRL-2014™, Lot. 70026474, American Type Culture Collection, USA) were cultured in Dulbecco's Modified Eagle Medium (Sigma-Aldrich, St. Louis, MO, USA) supplemented with 10% fetal bovine serum and penicillin-streptomycin (1X) at 37˚C and 5% $CO_2$ in a humidified atmosphere, and subcultured to a maximum of seven passages. HGF-1 cells were seeded at a density of 7,000 cells/cm² until grown to confluent, and then treated with test substances at the indicated concentrations and times. AG-1478 and TAPI-1 were dissolved in dimethyl sulfoxide (Sigma-Aldrich, St. Louis, MO, USA). S1PC was dissolved in Dulbecco's phosphate-buffered saline (-) (05913, Nissui Pharmaceutical Co., Tokyo, Japan).

## Extraction of total RNA and real-time quantitative PCR (qPCR) analysis

Total RNA was isolated from cells using RNAiso plus (Takara Bio Inc., Shiga, Japan). The isolated RNA was reverse-transcribed into complementary DNA with PrimeScript RT reagent kit with a genomic DNA Eraser (RR047A, Takara Bio Inc., Shiga, Japan) according to the manufacturer's instruction. An aliquot of complementary DNA was amplified on a CFX96 real-time PCR detection system (Bio-Rad Laboratories, Hercules, CA, USA) with KAPA SYBR fast qPCR master mix (KAPA Biosystems, Woburn, MA, USA). The reaction was run using the following program: 30 sec at 95˚C, 40 repeats of 10 sec at 95˚C, 10 sec at 63˚C, and 15 sec at 72˚C. The sequences of the specific primers (Integrated DNA Technologies, Inc., Coralville, IA) are listed in Table 1. The fold change in the relative mRNA level to β-actin was calculated based on the Ct (ΔΔCt) method [32].

## Western blotting

HGF-1 cells were lysed in a lysis buffer containing 150 mM NaCl, 1% Nonidet P-40 substitute (Nacalai Tesque, Kyoto, Japan), 0.5% sodium deoxycholate, 0.1% SDS, 50 mM Tris-HCl (pH 7.6), 1% Triton X-100, 5 mM EDTA, PhosSTOP™ (Sigma-Aldrich, St. Louis, MO, USA), and cOmplete™ protease inhibitor cocktail (Roche, Basel, Switzerland). The lysates were centrifuged at 14,500 rpm for 10 min at 4˚C, and then the supernatants were boiled in 4x SDS sample buffer containing 250 mM Tris-HCl (pH6.8), 8% SDS, 40% glycerol, 2% BPB, and 400 mM DTT for 5 min at 98˚C. The total protein amount in the supernatants was determined with the Pierce™ BCA Protein Assay kit (Thermo Fisher Scientific, Sunnyvale, CA, USA) with bovine serum albumin as a standard. Equal amounts of protein extracts (10 μg) were separated on SDS-polyacrylamide gels and transferred onto nitrocellulose membranes (Bio-Rad Laboratories, Hercules, CA, USA) or polyvinylidene difluoride membranes (Cytiva, Tokyo, Japan). The membranes were treated with a primary antibody against MMP-1 (1:500 dilution), EGFR and pEGFR (each

**Table 1. List of primers for real-time quantitative PCR.**

| Target | Sequence | | | |
|--------|----------|-----|---------------------------------|-----|
| *β-actin* | Forward | 5'- | CGC GAG AAG ATG ACC CAG AT | -3' |
| | Reverse | 5'- | GGT GAG GAT CTT CAT GAG GTA GTC | -3' |
| *MMP-1* | Forward | 5'- | ATG CTG AAA CCC TGA AGG TG | -3' |
| | Reverse | 5'- | CTG CTT GAC CCT CAG AGA CC | -3' |
| *MMP-2* | Forward | 5'- | AGG GCA CAT CCT ATG ACA GC | -3' |
| | Reverse | 5'- | ATT TGT TGC CCA GGA AAG TG | -3' |
| *MMP-3* | Forward | 5'- | GCA GTT TGC TCA GCC TAT CC | -3' |
| | Reverse | 5'- | GAG TGT CGG AGT CCA GCT TTC | -3' |
| *MMP-8* | Forward | 5'- | TCC AGC AAG AAC ATT TCT TCC | -3' |
| | Reverse | 5'- | CAG CCA TAT CTA CAG TTA AGC C | -3' |
| *MMP-13* | Forward | 5'- | CCT GGA GCA CTC ATG TTT CCT AT | -3' |
| | Reverse | 5'- | GAC TGG ATC CCT TGT ACA TCG TC | -3' |
| *MMP-14* | Forward | 5'- | CAT TGG AGG AGA CAC CCA CT | -3' |
| | Reverse | 5'- | TGG GGT TTT TGG GTT TAT CA | -3' |
| *TIMP-1* | Forward | 5'- | CTG TTC TTC CTG TGG CTG AT | -3' |
| | Reverse | 5'- | TCC GTC CAC AAG CAA TGA GT | -3' |
| *TIMP-2* | Forward | 5'- | CTG GAC GTT GGA GGA AAG AA | -3' |
| | Reverse | 5'- | GTC GAG AAA CTC CTG CTT GG | -3' |

at 1:1,000), or β-actin (1:2,000) overnight at 4˚C. Immunoreactive proteins were visualized with Armasham ECL Prime peroxidase solution (Cytiva, Tokyo, Japan) or Clarity Max Western ECL Substrate (Bio-Rad Laboratories, Hercules, CA, USA) by using ChemiDoc™ MP (Bio-Rad Laboratories, Hercules, CA, USA) after incubation with an HRP-conjugated secondary antibody (1:4,000) for 1 h at room temperature. The density of each immunoreactive band was quantitated with Image Lab™ software ver. 4.1 (Bio-Rad Laboratories, Hercules, CA, USA).

## Enzyme-Linked Immunosorbent Assay (ELISA)

Quantification of MMP-1 in culture media was performed using a human MMP-1 ELISA kit (ELH-MMP1-1, RayBiotech Inc., Norcross, GA, USA) according to the manufacturer's protocol. An aliquot of culture media (500 μL) was collected and centrifuged at 14,500 rpm for 10 min at 4˚C. The supernatants were stored at -80˚C until use. The colorimetric absorbance in each well was measured using Multiskan GO Microplate Spectrophotometer (Thermo Scientific, Vantaa, Finland) at a test wavelength of 450 nm. The concentration of MMP-1 protein in media was normalized to the amount of total cell protein in each well.

## Statistical analysis

Data analyses were performed using EZR Version 1.55 [33]. Data are expressed as mean ± standard deviation (S.D.). Statistical significance was assessed by one-way analysis of variance (ANOVA) followed by post hoc Bonferroni's multiple tests. Differences at $p < 0.05$ were considered statistically significant.

## Results

### S1PC suppresses LPS-induced MMP-1 expression and secretion

We first investigated whether S1PC affects the LPS-induced mRNA expressions of MMPs and TIMPs. HGF-1 cells were treated with S1PC (300 μM) in the presence or absence of LPS (3 μg/

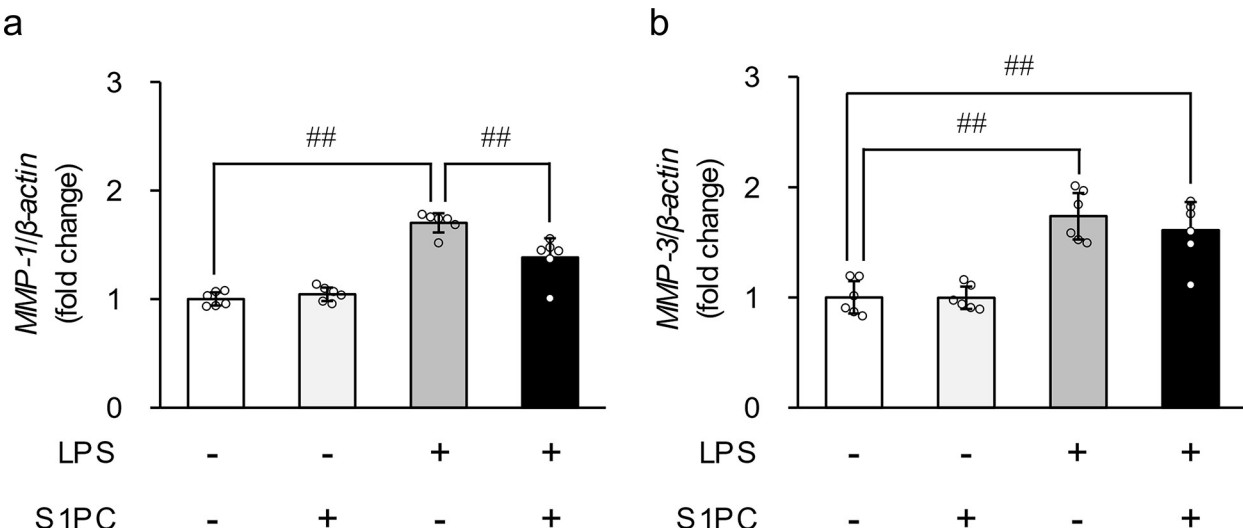

**Fig 1. The effect of S-1-propenyl-L-cysteine (S1PC) on the lipopolysaccharide (LPS)-induced gene expression of matrix metalloproteinase (MMP)-1 and -3.** HGF-1 cells were exposed to S1PC (300 μM) in the presence or absence of LPS (3 μg/mL) for 48 h. Bar graphs show the mRNA level of MMP-1 and -3, calculated relative to the level in each vehicle group. Values represent the mean ± S.D. (n = 5). ## $p < 0.01$.

mL) for 48 h. As shown in Fig 1, the gene expression levels of both MMP-1 and MMP-3 were significantly increased in LPS-treated cells. Simultaneous treatment with S1PC significantly suppressed the augmentation of MMP-1 mRNA level induced by LPS, whereas it did not affect the increase of MMP-3 mRNA level. The mRNA level of MMP-1 and MMP-3 in the cells treated with S1PC alone remained unchanged. Both LPS and S1PC did not affect the mRNA level of other MMPs and TIMP-1 and -2 (S1 Fig).

We then examined the concentration- and time-dependent effect of S1PC on the LPS-induced MMP-1 expression. Exposure of HGF-1 cells to LPS resulted in a significant increase in the mRNA level of MMP-1 at 24 and 48 h (Fig 2A). Co-treatment with S1PC significantly suppressed the LPS-induced mRNA level of MMP-1 at 48 h, but not at 24 h (Fig 2A). In addition, simultaneous treatment with S1PC (30, 100, or 300 μM) and LPS for 48 h resulted in a concentration-dependent decrease in the LPS-induced mRNA expression of MMP-1, and the significant effect was observed at 300 μM S1PC treatment (Fig 2B). Moreover, the intracellular protein level of MMP-1 in the cells co-treated with S1PC (300 μM) and LPS for 48 h was significantly lower than that treated with LPS alone (Fig 2C). Moreover, S1PC significantly reduced the increase in MMP-1 secretion elicited by LPS at 72 h (Fig 2D).

## S1PC suppresses LPS-induced transactivation of epidermal growth factor receptor (EGFR)

Epidermal growth factor (EGF) was reported to induce the expression of MMP-1 in primary human gingival fibroblasts [34]. In addition, LPS was shown to transactivate EGFR in *in vivo* [35, 36] and various cell lines such as rat small intestinal epithelial cell line IEC-6 [37] and human tongue squamous carcinoma cell line SCC-25 [38]. Thus, we examined whether S1PC prevents EGFR transactivation stimulated by LPS in HGF-1 cells. The phosphorylation level of EGFR was significantly augmented in LPS-treated cells at 5 min (Fig 3A). Simultaneous treatment with S1PC (300 μM) significantly suppressed the increase in the phosphorylation level of EGFR induced by LPS (Fig 3A). Since S1PC inhibited the LPS-induced phosphorylation of EGFR in HGF-1 cells, we next examined whether inhibition of EGFR phosphorylation

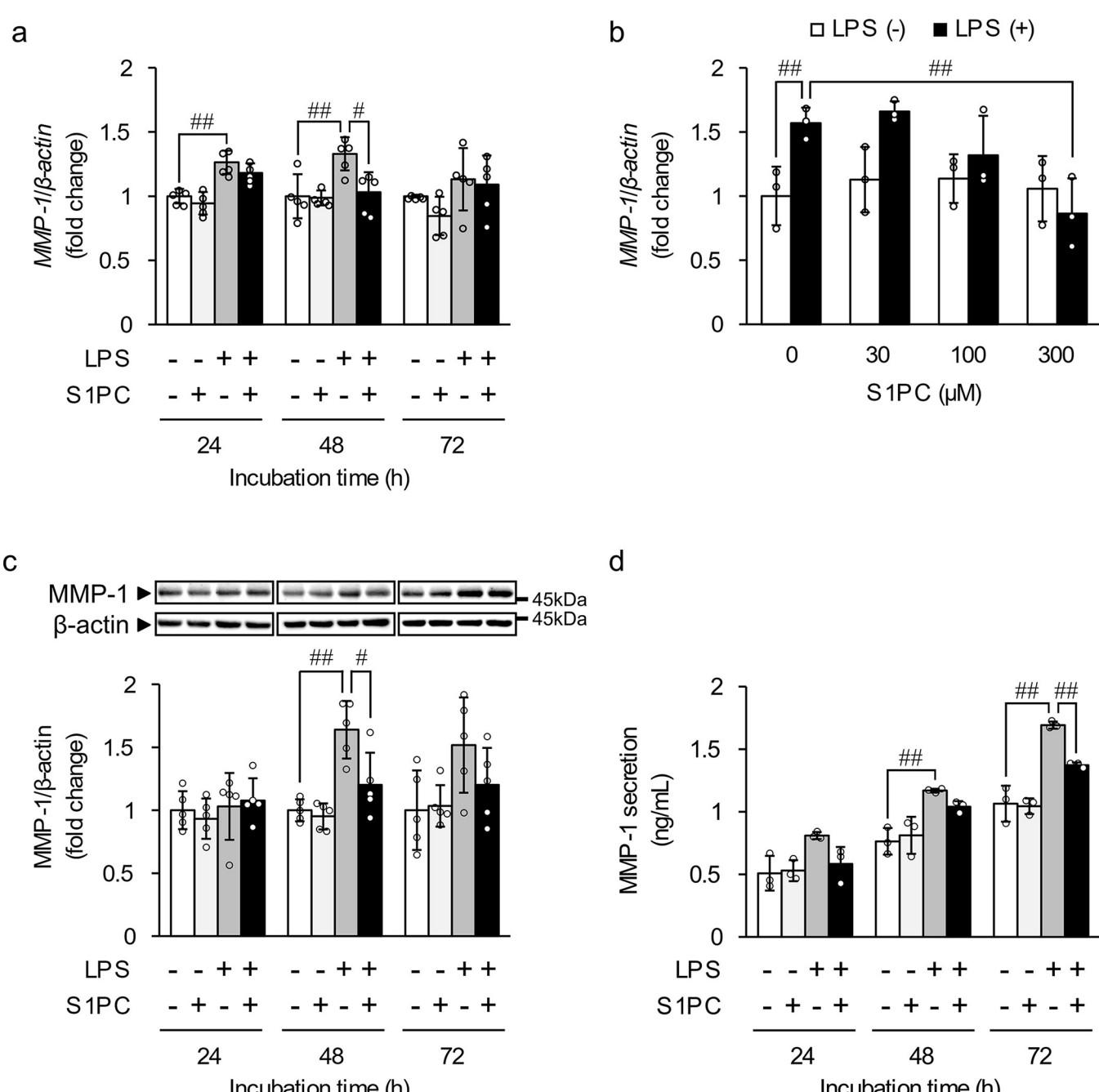

**Fig 2. The effect of S1PC on the LPS-induced expression and secretion of MMP-1.** (**a**) HGF-1 cells were exposed to S1PC (300 μM) in the presence or absence of LPS (3 μg/mL) for the indicated times. Bar graphs show the mRNA level of MMP-1, calculated relative to the level in the vehicle group at each time point. Values represent the mean ± S.D. (n = 5). [#] $p < 0.05$, [##] $p < 0.01$. (**b**) HGF-1 cells were exposed to S1PC in the presence or absence of LPS for 48 h at indicated concentrations. The graph shows the mRNA level of *MMP-1*. Values represent the mean ± S.D. (n = 3). [#] $p < 0.05$, [##] $p < 0.01$. (**c**) HGF-1 cells were exposed to S1PC (300 μM) in the presence or absence of LPS for the indicated times. Photographs show a representative result of Western blotting of MMP-1 with β-actin as an internal control. The graph shows the ratio of the MMP-1 band intensity relative to that of β-actin. Values represent the mean ± S.D. (n = 5). [#] $p < 0.05$, [##] $p < 0.01$. (**d**) HGF-1 cells were exposed to S1PC in the presence or absence of LPS for 72 h. The graph shows the concentration of MMP-1 in media determined by ELISA normalized to the total cell protein amount in each treatment group. Values represent the mean ± S.D. (n = 3). [##] $p < 0.01$.

a

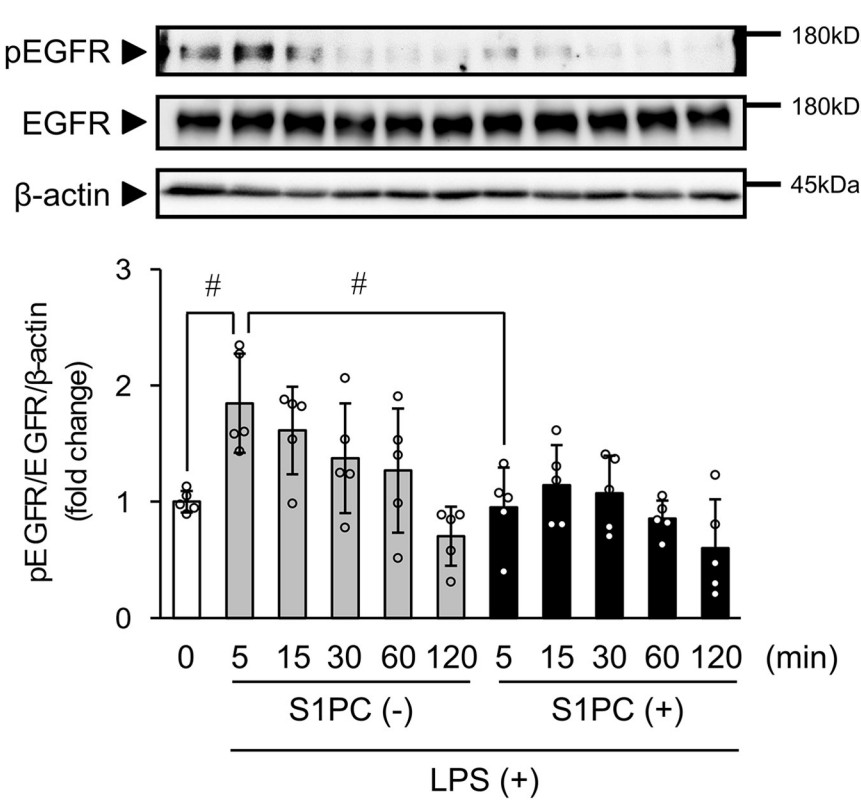

b

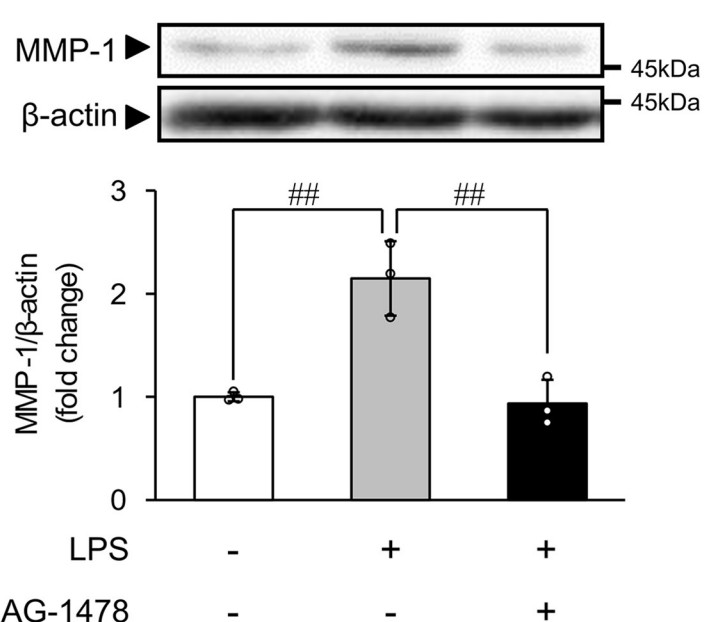

**Fig 3. The effect of S1PC on the LPS-induced epidermal growth factor receptor (EGFR) transactivation. (a)** HGF-1 cells were exposed to S1PC (300 μM) in the presence or absence of LPS (3 μg/mL) for the indicated times. Photographs show a representative result of Western blotting of phosphorylated EGFR (pEGFR) and EGFR with β-actin as an internal control. Bar graphs show the ratio of the pEGFR band intensity relative to that of EGFR and β-actin, calculated relative to the level at 0 min. Values represent the mean ± S.D. (n = 4). [##] $p < 0.01$. **(b)** HGF-1 cells were exposed to EGFR tyrosine kinase inhibitor AG-1478 (0.1 nM) in the presence or absence of LPS for 48 h. Photographs show a representative result of Western blotting of MMP-1 with β-actin as an internal control. Bar graphs show the ratio of the MMP-1 band intensity relative to that of β-actin. Values represent the mean ± S.D. (n = 3). [##] $p < 0.01$.

prevents LPS-induced enhancement of the intracellular MMP-1 protein level using an EGFR tyrosine kinase inhibitor AG-1478. As shown in Fig 3B, AG-1478 (0.1 nM) significantly suppressed the MMP-1 protein upregulation induced by LPS.

## S1PC suppresses LPS-induced tumor necrosis factor-α converting enzyme (TACE/ADAM17) cleavage

Previous studies showed that the activation of TACE is required for LPS-induced EGFR transactivation [39–41]. In addition, the processing at the proprotein convertase cleavage site is important for the functional activation of TACE [42]. We thus examined whether TACE activation can be detected by determining the protein level of cleaved TACE fragments using Western blot analysis. As shown in Fig 4A, it was found that exposure to LPS for 5 min significantly increased the level of cleaved TACE. Co-treatment with S1PC (300 μM) significantly suppressed the LPS-induced TACE cleavage at 5 min (Fig 4A). To assess the involvement of TACE cleavage in the increased protein level of MMP-1 by LPS in HGF-1 cells, we then investigated the effect of TAPI-1, an TACE inhibitor. As shown in Fig 4B, simultaneous treatment with TAPI-1 (1 μM) for 48 h significantly suppressed the LPS-induced increase in MMP-1 protein level.

## Discussion

Healthy human gingival connective tissue is mainly constituted of type I and III collagen fibers [7]. In particular, type I collagen fiber, rather than type III, is disorganized in diseased gingiva derived from patients with chronic gingivitis or periodontitis compared with healthy gingiva [43]. MMP-1 has a unique feature that specifically degrades native type I fibrillar collagen [44, 45]. Previous studies reported that the mRNA level of MMP-1, but not MMP-8, was increased in inflamed gingiva obtained from patients with severe gingivitis [12], and that the protein level of MMP-1 in gingival crevicular fluid decreased after phase I periodontal therapy [46]. Furthermore, exogenous interleukin-1β induces the degradation of type I collagen accompanied by the activation of MMP-1 and MMP-13 in human gingival fibroblasts [47]. These findings suggest that MMP-1 secreted from inflamed gingival fibroblasts contributes to the initiation of gingival breakdown in periodontal disease. Although AGE has been reported to alleviate probing pocket depth in patients with mild to moderate periodontitis [29], the mechanism underlying the therapeutic effect of AGE on gingival breakdown remains unknown. The present study demonstrated that S1PC, one of the beneficial bioactive substances in AGE, suppressed the LPS-induced expression and secretion of MMP-1 in HGF-1 cells, suggesting that the improvement of probing pocket depth by AGE is attributable, at least in part, to the suppressive effect of S1PC on MMP-1.

EGFR is a cell surface transmembrane receptor tyrosine kinase that regulates cell proliferation, survival, and differentiation [48]; it is also involved in proinflammatory responses such as spinal cord injury [49], ultraviolet irradiation [50], and atherosclerosis [51]. In addition,

a

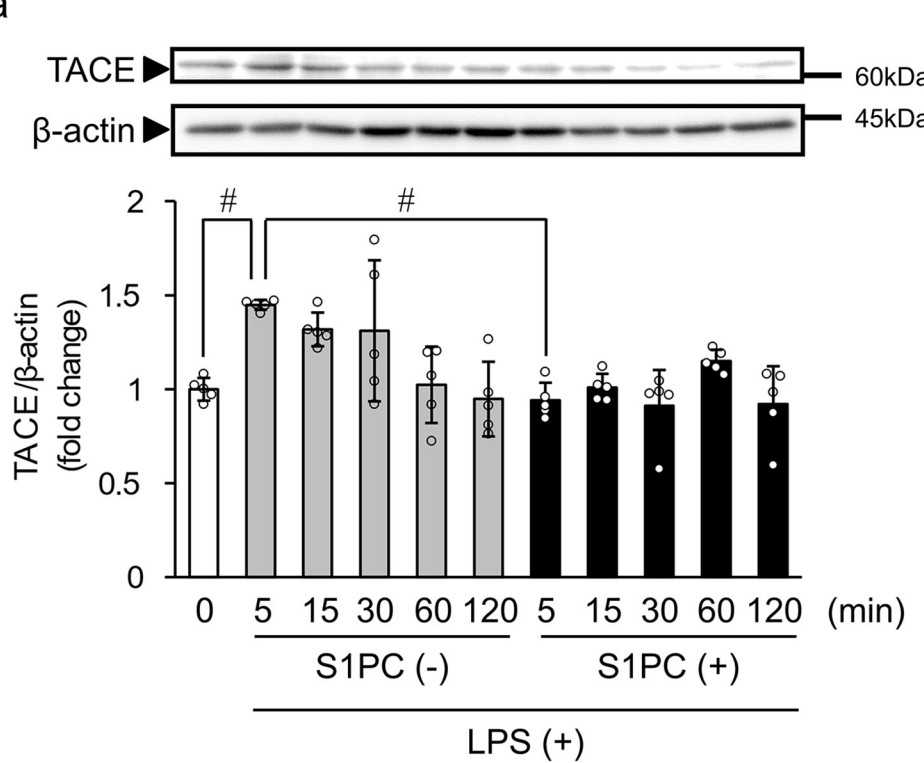

b

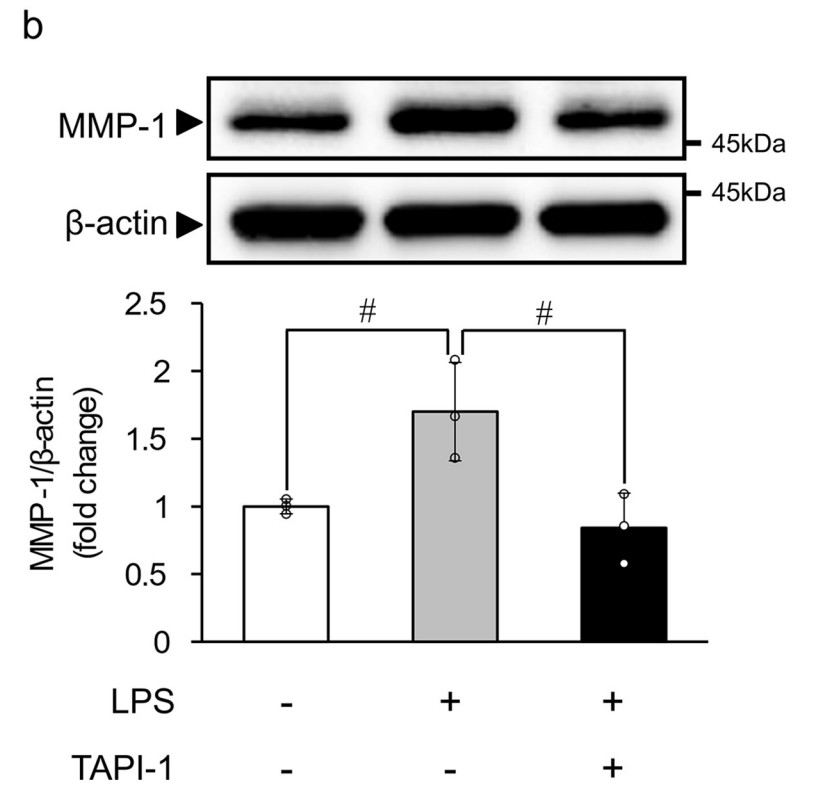

**Fig 4. The effect of S1PC on the LPS-induced tumor necrosis factor-α converting enzyme (TACE/ADAM17) cleavage.** (a) HGF-1 cells were exposed to S1PC (300 μM) in the presence or absence of LPS (3 μg/mL) for the indicated times. Photographs show a representative result of Western blotting of the active form of TACE with β-actin as an internal control. Bar graphs show the ratio of the active form of TACE band intensity relative to that of β-actin, calculated relative to the level at 0 min. Values represent the mean ± S.D. (n = 4). $^{\#\#}$ $p < 0.01$. (b) HGF-1 cells were exposed to an TACE inhibitor TAPI-1 (1 μM) in the presence or absence of LPS for 48 h. Photographs show a representative result of Western blotting of MMP-1. Bar graphs show the ratio of the MMP-1 band intensity relative to that of β-actin. Values represent the mean ± S.D. (n = 3). $^{\#}$ $p < 0.05$.

previous studies have shown that EGFR activation is also needed for LPS-induced expression of cyclooxygenase-2 in rat small intestinal epithelial cell line IEC-6 [37], and proinflammatory cytokine production and septic shock in LPS-treated mice [35, 36]. EGFR is highly expressed in epithelial and stromal cells of inflamed gingiva of patients with gingivitis, whereas no or low expression in healthy gingiva [52]. It has been reported that EGF upregulates the gene expression and protein level of MMP-1 in primary human gingival fibroblasts [34, 53]. Importantly, EGFR inhibitors, AG-1478 and gefitinib attenuated alveolar bone loss and inflammation in a ligature-induced experimental periodontal disease mouse model [54]. These studies suggest that EGFR inhibition is a potential target for the treatment of periodontal diseases. However, to our knowledge, it was unclear whether LPS can transactivate EGFR in gingival fibroblasts. In this study, we demonstrated that LPS induced the phosphorylation of EGFR, and that co-treatment with an EGFR tyrosine kinase inhibitor AG-1478 inhibited the increase in LPS-induced protein level of MMP-1 in HGF-1 cells. These results suggest that LPS would induce the expression of MMP-1 through the transactivation of EGFR in gingival fibroblasts. S1PC also prevented LPS-induced phosphorylation of EGFR, suggesting that it suppresses LPS-induced expression of MMP-1 through the blockade of EGFR transactivation in HGF-1 cells.

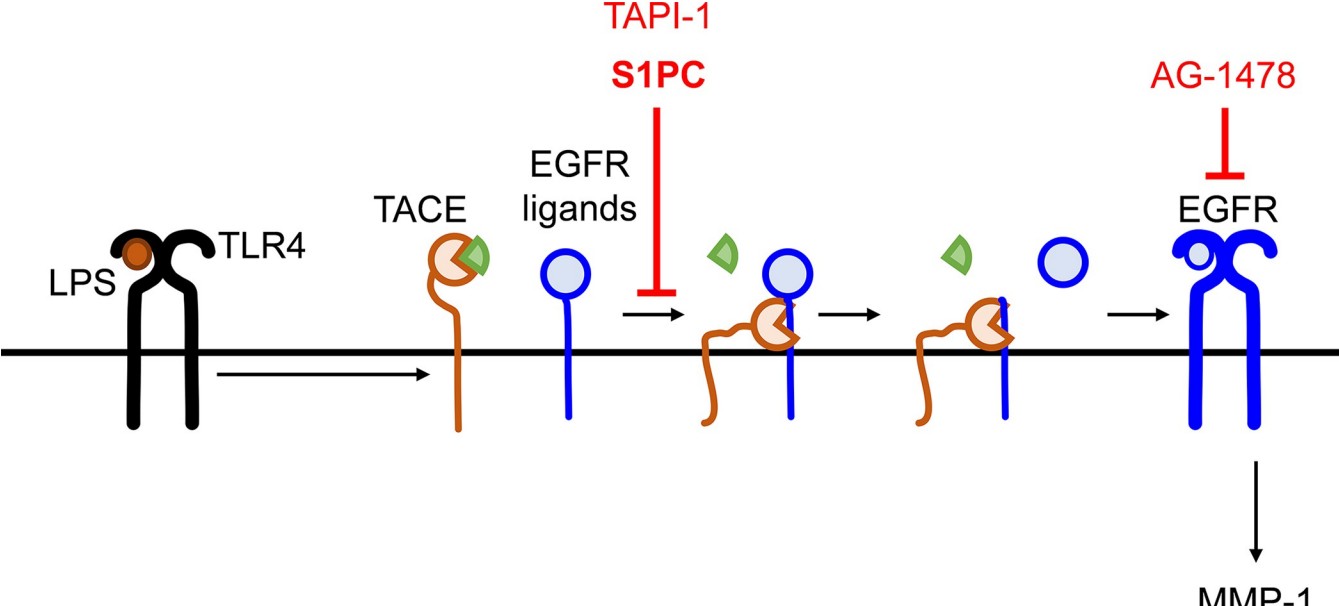

**Fig 5. A proposed mechanism by which S1PC suppressed the LPS-induced MMP-1 expression in human gingival fibroblasts.** The binding of LPS to TLR4 activates TACE and this activation promotes the phosphorylation of EGFR to induce the expression of MMP-1 in HGF-1 cells. Both EGFR tyrosine kinase and TACE inhibitors, AG-1478 and TAPI-1, antagonize the LPS-induced expression of MMP-1. S1PC suppresses the LPS-induced activation of TACE and subsequent phosphorylation of EGFR, thus ultimately suppressing the expression of MMP-1. TACE: tumor necrosis factor-α converting enzyme, EGFR: epidermal growth factor receptor, LPS: lipopolysaccharide, MMP-1: matrix metalloproteinase-1, S1PC: S-1-propenyl-L-cysteine, TLR4: toll-like receptor 4.

TACE is a member of transmembrane metalloproteinases involved in the shedding of various cell-surface proteins such as TNF-α [55] and EGFR ligands [56]. The protein level of TACE in the human gingival tissue and gingival crevicular fluid has been reported to positively correlate with the severity of periodontal disease [57, 58]. TACE-dependent EGFR transactivation is required for LPS-induced inflammatory responses in various cells such as human lung epithelial A549 cells [39], human cholangiocyte cells [40], and human tracheal smooth muscle cells [41]. The present study demonstrated that LPS increased the protein level of the active form of TACE, and that simultaneous treatment with TACE inhibitor TAPI-1 and LPS suppressed the increased LPS-induced MMP-1 protein level as did S1PC and an EGFR tyrosine kinase inhibitor AG-1478, suggesting that the activation of TACE is required for LPS-induced expression of MMP-1 in HGF-1 cells. In addition, we found that S1PC inhibited the increase in the active form of TACE induced by LPS. These results suggest that the suppressive effect of S1PC on the LPS-induced activation of TACE is involved in the blockade of EGFR transactivation with subsequent increased expression of MMP-1 in HGF-1 cells.

## Conclusion

In conclusion, we demonstrated for the first time that *P. gingivalis*-derived LPS induces the expression of MMP-1 through the TACE-EGFR axis in HGF-1 cells, and that S1PC, one of the beneficial bioactive substances in AGE, suppresses this induction possibly through blockade of TACE activation as summarized in Fig 5. Although further *in vivo* studies will be required to verify these *in vitro* findings in experimental animal models and patients with periodontal diseases, this novel action of S1PC may help the clinical management of patients with periodontal disease. Our present findings also would clarify the therapeutic mechanism by which intake of AGE alleviated periodontal disease in the patients with mild to moderate gingivitis and periodontitis [27, 29].

## Supporting information

**S1 Fig. The effect of *S*-1-propenyl-L-cysteine (S1PC) on the lipopolysaccharide (LPS)-induced gene expression of matrix metalloproteinases (MMPs) and tissue inhibitor of metalloproteinases (TIMPs).** HGF-1 cells were exposed to S1PC (300 μM) in the presence or absence of LPS (3 μg/mL) for 48 h. Bar graphs show the mRNA level of MMP-2, -8, -13, and -14, and TIMP-1 and -2, calculated relative to the level in each vehicle group. Values represent the mean ± S.D. (n = 5).
(TIF)

**S1 Raw images. Full-width of the membranes of the original blots used in the manuscript.** The contrast and brightness of these blots are modulated in the manuscript.
(PDF)

**S1 Dataset.**
(XLSX)

## Acknowledgments

The authors would like to thank all the members of our laboratories, especially Dr. Takami Oka for his helpful advice, encouragement, and critical reading of the manuscript.

## Author Contributions

**Conceptualization:** Hiroshi Nango, Masahiro Ohtani.

**Data curation:** Hiroshi Nango.

**Formal analysis:** Hiroshi Nango.

**Investigation:** Hiroshi Nango.

**Methodology:** Hiroshi Nango.

**Project administration:** Hiroshi Nango.

**Supervision:** Masahiro Ohtani.

**Validation:** Hiroshi Nango.

**Visualization:** Hiroshi Nango.

**Writing – original draft:** Hiroshi Nango.

**Writing – review & editing:** Masahiro Ohtani.

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
