## [Decision Letter · Decision Letter 0]

9 Feb 2023

PONE-D-22-27612S-1-propenyl-L-cysteine suppresses lipopolysaccharide-induced expression of matrix metalloproteinase-1 through inhibition of tumor necrosis factor-α converting enzyme-epidermal growth factor receptor axis in human gingival fibroblastsPLOS ONE

Dear Dr. Nango,

Thank you for submitting your manuscript to PLOS ONE. After careful consideration, we feel that it has merit but does not fully meet PLOS ONE’s publication criteria as it currently stands. Therefore, we invite you to submit a revised version of the manuscript that addresses the points raised during the review process.

We look forward to receiving your revised manuscript.

Kind regards,

Bruno Giros, Ph.D.

Academic Editor

PLOS ONE

Journal Requirements:

3. Thank you for providing the following Funding Statement:  

This study was funded by Wakunaga Pharmaceutical Co., Ltd.

The company has no financial interest in this publication. 

We note that one or more of the authors is affiliated with the funding organization, indicating the funder may have had some role in the design, data collection, analysis or preparation of your manuscript for publication; in other words, the funder played an indirect role through the participation of the co-authors. 

If the funding organization did not play a role in the study design, data collection and analysis, decision to publish, or preparation of the manuscript and only provided financial support in the form of authors' salaries and/or research materials, please review your statements relating to the author contributions, and ensure you have specifically and accurately indicated the role(s) that these authors had in your study in the Author Contributions section of the online submission form. Please make any necessary amendments directly within this section of the online submission form.  Please also update your Funding Statement to include the following statement: “The funder provided support in the form of salaries for authors [insert relevant initials], but did not have any additional role in the study design, data collection and analysis, decision to publish, or preparation of the manuscript. The specific roles of these authors are articulated in the ‘author contributions’ section.

Reviewers' comments:

Reviewer's Responses to Questions

**Comments to the Author**

1. Is the manuscript technically sound, and do the data support the conclusions?

Reviewer #1: Yes

Reviewer #2: Partly

2. Has the statistical analysis been performed appropriately and rigorously? 

Reviewer #1: Yes

Reviewer #2: Yes

3. Have the authors made all data underlying the findings in their manuscript fully available?

Reviewer #1: Yes

Reviewer #2: Yes

4. Is the manuscript presented in an intelligible fashion and written in standard English?

Reviewer #1: Yes

Reviewer #2: Yes

5. Review Comments to the Author

Reviewer #1: The introduction part presents a good introduction to the topic and the literature is relevant. The research is methodologically correctly appointed, and the presentation of the results (tables, figures) corresponds to the objectives of the research and the set hypothesis. The discussion is extensive and debates the set goal and results of the study. Altogether, the paper is well designed and well written.

Reviewer #2: The manuscript by Hango. Et al. describes the impact of S1PC (a bioactive compound derived from aged garlic extract) on HGF1 cells. The authors observed decreased mRNA expression, intracellular levels, and secretion of MMP1 after the treatment. They linked this reduction with the inhibition of LPS-induced phosphorylation of EGFR and cleavage of TACE. Overall, the manuscript depicts a straightforward hypothesis based on the group's extensive experience with aged garlic extracts using a simple experimental design.

Major issues: Mostly related to the explanation of sample size and technical replicas. In all figures, the number of replicas is relatively low (n=3-5), and it is unclear if that number depicts separate experiments or just replicas within a single experiment. Further, the low number of replicas impacted the statistical differences between groups with clear tendencies that are not significant (i.e., Figure 2a at 72h, figure 2b at 30uM between no LPS and LPS, and so on). Given the experimental design’s simplicity (not primary cell culture, but a cell line), the number of replicas should be high enough to guarantee a precise statistical analysis. Also, I suggest using all bar graphs the individual values to understand high SD.

Minor issues: Since 1998 (ref 2), a significant understanding of the pathogenesis of periodontal disease has been achieved in a way that is no longer considered a “chronic infection,” as stated in the abstract, or caused by pathogenic bacteria (line 29). Instead, Periodontitis is a chronic multifactorial inflammatory disease associated with dysbiotic plaque biofilms (https://doi.org/10.1002/JPER.17-0721).

Figure 1 b, has a duplicated group (LPS+, S1PC+).

6. PLOS authors have the option to publish the peer review history of their article (what does this mean?). If published, this will include your full peer review and any attached files.

Reviewer #1: No

Reviewer #2: No

---

## [Author Response · Author response to Decision Letter 0]

16 Feb 2023

Point-by-point responses to the Editor and Reviewers (Manuscript ID: PONE-D-22-27612) 

We are grateful to the Editor and Reviewers for evaluating our manuscript and for providing us with such valuable comments. Below, please find our point-by-point responses to each comment; the comments made by the reviewer are in black font, and our responses are in blue font. The revised areas in the manuscript are marked with red text. 

Reviewer #1

The introduction part presents a good introduction to the topic and the literature is relevant. The research is methodologically correctly appointed, and the presentation of the results (tables, figures) corresponds to the objectives of the research and the set hypothesis. The discussion is extensive and debates the set goal and results of the study. Altogether, the paper is well designed and well written.

Response: We appreciate your kind comment. 

Reviewer #2:

The manuscript by Hango. Et al. describes the impact of S1PC (a bioactive compound derived from aged garlic extract) on HGF1 cells. The authors observed decreased mRNA expression, intracellular levels, and secretion of MMP1 after the treatment. They linked this reduction with the inhibition of LPS-induced phosphorylation of EGFR and cleavage of TACE. Overall, the manuscript depicts a straightforward hypothesis based on the group's extensive experience with aged garlic extracts using a simple experimental design.

Response: We appreciate your kind comment and hope that we were able to adequately address all your concerns. Your comments and suggestions have greatly improved our manuscript.

Major issues: Mostly related to the explanation of sample size and technical replicas. In all figures, the number of replicas is relatively low (n=3-5), and it is unclear if that number depicts separate experiments or just replicas within a single experiment.

Response: We have missed to clarify details for the replica. The number of replicates in each figure caption represents the number of independent experiments. We have added details about the number of replicates to each figure caption in the revised manuscript. In addition, we are sorry to have made careless mistakes for the number of experiments. The incorrect sample sizes in Fig. 1, 3a, 4a, and Fig. S1 in the original manuscript have been corrected in the revised manuscript.

Further, the low number of replicas impacted the statistical differences between groups with clear tendencies that are not significant (i.e., Figure 2a at 72h, figure 2b at 30uM between no LPS and LPS, and so on). Given the experimental design’s simplicity (not primary cell culture, but a cell line), the number of replicas should be high enough to guarantee a precise statistical analysis. 

Response: We agree that this point is important from a statistical viewpoint. As the reviewer pointed out, it has been reported that the statistical power increases with an increase in sample size (Naegle K et al., Sci. Signal., 2015), so we expect statistically significant differences in our clear trend data with increasing sample size. However, we consider, at least, that our data sufficiently support the beneficial effects of S1PC with statistical significance (LPS alone vs. LPS + S1PC) even at the current sample size. We also think that our sample size (n = 3－6) is not particularly low in general, considering that several articles using cell lines recently published in PLOS ONE were conducted with two to four independent experiments (Maniscalco E et al., Plos One, 2023; Sharif M et al., Plos One, 2023; Shi Y et al., Plos One, 2023). This suggestion is controversial as a general issue of determining an appropriate sample size in in vitro experiments and should be given proper attention in future experiments.

Also, I suggest using all bar graphs the individual values to understand high SD.

Response: Overall, we agree with you on this point. We added individual values as dot plots to each graph in the revised manuscript.

Minor issues: Since 1998 (ref 2), a significant understanding of the pathogenesis of periodontal disease has been achieved in a way that is no longer considered a “chronic infection,” as stated in the abstract, or caused by pathogenic bacteria (line 29). Instead, Periodontitis is a chronic multifactorial inflammatory disease associated with dysbiotic plaque biofilms (https://doi.org/10.1002/JPER.17-0721).

Response: Thank you for your important suggestion. We have modified these sentences (line 2－3 in the abstract section and line 27－29 in the introduction section) and replaced reference 2 to the following suggested reference in the revised manuscript.

2. Papapanou PN, Sanz M, Buduneli N, Dietrich T, Feres M, Fine DH, et al. Periodontitis: Consensus report of workgroup 2 of the 2017 World Workshop on the Classification of Periodontal and Peri-Implant Diseases and Conditions. J Periodontol. 2018;89: S173–S182. doi:10.1002/JPER.17-0721

Figure 1 b, has a duplicated group (LPS+, S1PC+).

Response: Done as suggested.

---

## [Decision Letter · Decision Letter 1]

6 Apr 2023

S-1-propenyl-L-cysteine suppresses lipopolysaccharide-induced expression of matrix metalloproteinase-1 through inhibition of tumor necrosis factor-α converting enzyme-epidermal growth factor receptor axis in human gingival fibroblasts

PONE-D-22-27612R1

Dear Dr. Nango,

We’re pleased to inform you that your manuscript has been judged scientifically suitable for publication and will be formally accepted for publication once it meets all outstanding technical requirements.

Kind regards,

Bruno Giros, Ph.D.

Academic Editor

PLOS ONE

Additional Editor Comments (optional):

Reviewers' comments:

Reviewer's Responses to Questions

**Comments to the Author**

1. If the authors have adequately addressed your comments raised in a previous round of review and you feel that this manuscript is now acceptable for publication, you may indicate that here to bypass the “Comments to the Author” section, enter your conflict of interest statement in the “Confidential to Editor” section, and submit your "Accept" recommendation.

Reviewer #2: All comments have been addressed

2. Is the manuscript technically sound, and do the data support the conclusions?

Reviewer #2: Yes

3. Has the statistical analysis been performed appropriately and rigorously? 

Reviewer #2: Yes

4. Have the authors made all data underlying the findings in their manuscript fully available?

Reviewer #2: Yes

5. Is the manuscript presented in an intelligible fashion and written in standard English?

Reviewer #2: Yes

6. Review Comments to the Author

Reviewer #2: All comments have been satisfactory addressed.

The author corrected the minor issues and specified the number of independent experiments performed.

7. PLOS authors have the option to publish the peer review history of their article (what does this mean?). If published, this will include your full peer review and any attached files.

Reviewer #2: No

---

## [Editor Report · Acceptance letter]

13 Apr 2023

PONE-D-22-27612R1 

*S*-1-propenyl-L-cysteine suppresses lipopolysaccharide-induced expression of matrix metalloproteinase-1 through inhibition of tumor necrosis factor-α converting enzyme-epidermal growth factor receptor axis in human gingival fibroblasts 

Dear Dr. Nango:

I'm pleased to inform you that your manuscript has been deemed suitable for publication in PLOS ONE. Congratulations! Your manuscript is now with our production department. 

Kind regards, 

on behalf of

Professor Bruno Giros 

Academic Editor

PLOS ONE